# Prospective evaluation of percutaneous hepatic perfusion with melphalan as a treatment for unresectable liver metastases from colorectal cancer

T. Susanna Meijer[1]*, Jan H. N. Dieters[1], Eleonora M. de Leede[2], Lioe-Fee de Geus-Oei[1,3], Jaap Vuijk[4], Christian H. Martini[4], Arian R. van Erkel[1], Jacob Lutjeboer[1], Rutger W. van der Meer[1], Fred G. J. Tijl[5], Ellen Kapiteijn[6], Alexander L. Vahrmeijer[2], Mark C. Burgmans[1]

1 Department of Radiology, Leiden University Medical Center, Leiden, The Netherlands, 2 Department of Surgery, Leiden University Medical Center, Leiden, The Netherlands, 3 Biomedical Photonic Imaging Group, University of Twente, Enschede, The Netherlands, 4 Department of Anesthesiology, Leiden University Medical Center, Leiden, The Netherlands, 5 Department of Extra Corporal Circulation, Leiden University Medical Center, Leiden, The Netherlands, 6 Department of Medical Oncology, Leiden University Medical Center, Leiden, The Netherlands

* t.s.meijer@lumc.nl

**Data Availability Statement:** All relevant data are within the paper and its Supporting Information files.

## Abstract

### Purpose

Percutaneous hepatic perfusion with melphalan (M-PHP) is increasingly used in patients with liver metastases from various primary tumors, yet data on colorectal liver metastases (CRLM) are limited. The aim of this study was to prospectively evaluate the efficacy and safety of M-PHP in patients with CRLM.

### Materials and methods

Prospective, single-center, single-arm phase II study of M-PHP with hemofiltration in patients with unresectable CRLM. Proven, extrahepatic metastatic disease was one of the exclusion criteria. Primary outcomes were overall response rate (ORR) and best overall response (BOR). Secondary outcomes were overall survival (OS), progression-free survival (PFS), hepatic PFS (hPFS), and safety.

### Results

A total of 14 M-PHP procedures were performed in eight patients between March 2014 and December 2015. All patients (median age 56 years, ranging from 46 to 68) had received (extensive) systemic chemotherapy before entering the study. The ORR was 25.0%, with two out of eight patients showing partial response as BOR. Median OS was 17.3 months (ranging from 2.6 to 30.9) with a one-year OS of 50.0%. Median PFS and hPFS were 4.4 and 4.5 months, respectively. No serious adverse events occurred. Grade 3/4 hematologic

**Funding:** YES - The Leiden University Medical Center received financial support and in kind contributions from Delcath Systems Inc. for conducting studies on M-PHP. No grant number applies. The sponsor did not play any rol in the study design, data collection and analysis, decision to publish, or preparation of the manuscript.

**Competing interests:** The Leiden University Medical Center received financial support and in kind contributions from Delcath Systems Inc. for conducting studies on M-PHP. No grant number applies. On behalf of all authors, I declare that no competing interests exist. This does therefore not alter our adherence to PLOS ONE policies on sharing data and materials.

**Abbreviations:** BOR, Best overall response; CFV, Common femoral vein; CR, Complete response; CRC, Colorectal cancer; CRLM, Colorectal liver metastases; CTCAE v4.03, Common terminology criteria for adverse events version 4.03; GEN 2, Second-generation; hPFS, hepatic Progression-free survival; IVC, Inferior vena cava; M-PHP, Percutaneous hepatic perfusion with melphalan; ORR, Overall response rate; OS, Overall survival; PD, Progressive disease; PFS, Progression-free survival; PR, Partial response; SD, Stable disease; RECIST, Response Evaluation Criteria in Solid Tumors.

adverse events were observed in the majority of patients, though all were transient and well-manageable.

## Conclusion

M-PHP is a safe procedure with only limited efficacy in patients with unresectable CRLM who already showed progression of disease after receiving one or more systemic treatment regimens.

## Introduction

Colorectal cancer (CRC) is the third most common cancer type worldwide. In 2018, approximately 1.8 million new cases of CRC were diagnosed, which accounted for 9.2% (880.000) of all cancer-related deaths [1]. The most common site of distant metastases in CRC is the liver as the majority of venous blood from the colon drains into the hepatic portal vein via the superior and inferior mesenteric veins. Around 15–25% of patients will present with synchronous colorectal liver metastases (CRLM), and ultimately approximately 50% of patients with CRC will develop liver metastases at some point in the course of their disease [2,3].

Surgical resection is considered standard of care for patients with resectable CRLM with a median overall survival (OS) ranging from 36–56 months [4–6]. Despite the improvement of surgical techniques, expansion of the indications for surgery, and advances in neoadjuvant therapies, only about 25% of patients is eligible for surgery at the time of diagnosis [7]. In patients with unresectable CRLM, systemic therapy is considered to be the first treatment modality with a reported median OS of approximately 2.5 years [8]. Liver-directed therapies such as radioembolization, chemoembolization, hepatic arterial infusion pump chemotherapy, or isolated hepatic perfusion (IHP) may offer an alternative treatment with limited systemic side-effects, but are generally not considered as first-line therapy in patients with CRLM.

Percutaneous hepatic perfusion with melphalan (M-PHP) is a novel therapy that was developed as minimally invasive alternative to IHP. IHP is an invasive, complex surgical procedure in which the liver is isolated from the systemic circulation followed by infusion of a high dose of chemotherapy into the common hepatic artery and/or portal vein [9–13]. In patients with CRLM, hepatic response rates of 50–59% and a median OS of 24.8–28.8 months have been reported after IHP with a high-dose of melphalan [14,15]. A major drawback of IHP is that it is not repeatable and associated with considerable morbidity and mortality rates up to 7% as a result of the invasive surgical procedure [14,15].

M-PHP is a repeatable, well-tolerated procedure with an acceptable safety profile [16–18] that is able to prolong progression-free survival in patients with liver metastases from ocular melanoma [17,19–22]. Up to now, data on M-PHP in CRLM remain limited. Only a small number of patients with CRLM have been studied while they were part of a heterogeneous cohort of patients with liver metastases from different primary tumors [23–25]. Moreover, these studies did not report tumor response and survival in CRLM patients. The aim of this study was to prospectively evaluate the efficacy and safety of M-PHP in patients with unresectable CRLM.

## Materials and methods

### Patients selection and study design

The current study was designed as a prospective, single-arm, single-center phase II study and registered in advance at www.trialregister.nl (NTR4050). Ethical approval was obtained from

the local ethics committee (Leiden University Medical Center) and the study was conducted in accordance with the Declaration of Helsinki, 2013 version. Written informed consent was obtained from all patients before inclusion.

Patients with histologically proven and unresectable CRLM were eligible for the study. Exclusion criteria are listed in Table 1. Prior to inclusion, all patients were discussed at a multi-disciplinary team (MDT) meeting. To achieve an acceptable inclusion rate few restrictive exclusion criteria were incorporated in the study protocol. Patients with unresectable CRLM were eligible for inclusion regardless of any prior systemic treatment. This allowed inclusion of patients who were intolerant to systemic chemotherapy or chemo-naïve and unwilling to undergo systemic therapy. However, our MDT always gave preference to first-line systemic chemotherapy over study inclusion in chemo-naïve patients.

Treatment consisted of two M-PHP procedures at a 5–8 weeks interval. Patients with progressive disease (PD) or unacceptable adverse events (AEs) after the first M-PHP procedure, did not receive a second procedure. The melphalan dose was reduced with 20–25% if patients developed grade 3/4 hematologic toxicity after the first procedure. All patients received a subcutaneous injection of granulocyte-colony stimulating factor (pegfilgrastim 6 mg) within 72h after each M-PHP. M-PHP was scheduled at least one month after resection of the primary tumor to prevent gastro-intestinal bleeding complications as a result of per-procedural heparinization.

## Procedure details

The M-PHP procedure has been described in greater detail previously [26]. Essential steps are discussed below.

Hepatic angiography was performed approximately one week prior to the first M-PHP and hepatico-enteric anastomoses were embolized if deemed necessary to prevent inadvertent flow of melphalan to the gastrointestinal tract.

All M-PHP procedures were performed under general anesthesia with continuously monitoring of vital signs. Per-procedural heparin was administered to achieve an activated clotting

**Table 1. Exclusion criteria.**

| Laboratory test results | Other |
|---|---|
| APTT > 1.5 × ULN | Age < 18 or > 75 years |
| PT > 1.5 × ULN | Extrahepatic metastatic disease (on CECT or FDG-PET/CT) |
| Leukocytes < 3.0 × 10$^9$/L | WHO performance status ≥ 2 |
| Thrombocytes < 100 × 10$^9$/L | Severe comorbidity[a] |
| Creatinine clearance < 40 ml/min | < 40% healthy liver tissue |
| AST > 2.5 × ULN | Vascular anatomy impeding M-PHP |
| ALT > 2.5 × ULN | Prior Whipple's surgery |
| Serum bilirubin > 1.5 × ULN | Intracranial lesions with propensity to bleed (on CT/MRI) |
| ALP > 2.5 × ULN | Pregnancy |

ALP, alkaline phosphatase; ALT, alanine aminotransferase; APTT, activated partial thromboplastin time; AST, aspartate aminotransferase; CECT, contrast-enhanced computed tomography of chest and abdomen; FDG-PET/CT, positron emission tomography with integrated non-contrast enhanced computed tomography and 18F-2-fluoro-2-deoxy-D-glucose as radiotracer; M-PHP, percutaneous hepatic perfusion with melphalan; PT, prothrombin time; ULN, upper limit of normal.

[a] e.g. cardiovascular or pulmonary disease precluding general anaesthesia, diabetes with nephropathy, active infections, other liver disease.

time of ≥ 450 seconds. A 18-F sheath was placed percutaneously into the right common femoral vein (CFV) and through the sheath a 16-F double-balloon catheter (Isofuse Isolation Aspiration Catheter, Delcath Systems Inc, New York, NY, USA) was placed in the inferior vena cava (IVC) via the right CFV. The cranial balloon was inflated at the atriocaval junction and the caudal balloon at the infrahepatic portion of the IVC to prevent flow of melphalan into the systemic circulation. Melphalan 3 mg/kg (maximum dose 220 mg) was infused into the proper hepatic artery. Alternatively, the dose was split and infused into both the right and left hepatic artery. The chemosaturated blood was then aspirated through catheter fenestrations in a segment between the two balloons, pumped through an extracorporeal hemofiltration system (GEN 2 CHEMOSAT® filtration system, Delcath Systems Inc, New York, NY, USA) and returned to the patient via a 10-F sheath in the right internal jugular vein. Extracorporeal filtration was continued for 30 minutes after completion of melphalan infusion. Protamine sulphate 3 mg/kg was administrated at the end of the procedure. The arterial sheath in the left common femoral artery was removed and hemostasis was achieved using a closure device.

## Follow-up

Contrast-enhanced computed tomography (CECT) of the chest and abdomen was performed at baseline, 4–8 weeks after each M-PHP, and then every 3 months in the first year and every 6 months thereafter until progression occurred.

Blood tests were performed daily during hospital admission and at several fixed time points after discharge. Adverse events were continuously monitored and reported according to the Common Terminology Criteria for Adverse Events version 4.03 (CTCAE v4.03).

## Endpoints

All images were reviewed by independent radiologists using the Response Evaluation Criteria in Solid Tumors (RECIST) 1.1 criteria [27]. Primary endpoint was the overall response rate (ORR). ORR was defined as the percentage of patients with complete response (CR) or partial response (PR). Best overall response (BOR) was used to determine ORR. BOR was defined as the best response at any time point after the first M-PHP and prior to the start of any other anti-cancer therapy. Secondary endpoints included best hepatic response, OS, progression-free survival (PFS), hepatic PFS (hPFS), and safety.

## Statistical analyses

OS was defined as time of first M-PHP until death due to any cause or censoring. PFS was defined as time of first M-PHP until progressive disease (PD), death due to any cause or censoring. hPFS was defined as time of first M-PHP until progression of liver disease, death due to any cause, or censoring.

The sample size calculations were based on the primary outcome measure of response after two M-PHP procedures. In a previous clinical trial where patients were treated with one IHP procedure, an ORR was observed in over 50% of patients [14]. Treatment with two M-PHP perfusions was expected to increase this response percentage. We choose a sample size that allows the response percentage to be determined with sufficient accuracy, i.e. with a sufficiently narrow confidence interval. Assuming a true response percentage of 60%, a sample size of 34 patients will yield a two-sided confidence interval of length 0.33 (± 16.5% around the observed proportion).

Median OS, median PFS, and hPFS were measured in days and subsequently converted into months. Graphs shown in this study were generated with dedicated software (SPSS 23.0, SPSS Inc., Chicago, IL, USA) using the Kaplan-Meier method.

## Results

Our study was terminated prematurely due to slow recruitment. Several factors contributed to the slow recruitment, such as the availability of alternative therapies (e.g. systemic chemotherapy or radioembolization), competing trials with systemic drugs or intra-arterial therapies, and ineligibility of screened patients.

Between March 2014 and December 2015, a total of 16 patients with unresectable CRLM were assessed for eligibility (Fig 1). Of these, eight patients were excluded for the following reasons: extrahepatic disease ($n = 4$) or because systemic chemotherapy was preferred during the MDT meeting as standard first-line therapy ($n = 4$). Thus, eight patients with a median age of 56 years (range 46–68) participated in this study. Baseline patient characteristics are summarized in Table 2. All patients received some form of systemic therapy before entering the study and half of the patients received prior surgical resection of CRLM. The median interval between diagnosis of CRLM and first M-PHP was 23.7 months (ranging from 8.3 to 35.1).

A total of 14 M-PHPs were performed in eight patients. Six patients received two M-PHP procedures as per protocol and the other two patients received only one M-PHP procedure due to PD after the first procedure. Median melphalan dose was 220 mg (ranging from 190 to 220) for the first cycle and 220 mg (ranging from range 160 to 220) for the second cycle. In all procedures, hospital length-of-stay was two or three days.

There was no loss to follow-up. Tumor response and survival outcomes are reported in Table 3. ORR was 25% with two out of eight patients showing partial response (PR) (Fig 2). Three patients (38%) showed stable disease (SD) as BOR and the other three patients (38%) showed PD.

At the time of study termination, all patients had passed away. Median OS was 17.3 months (ranging from 2.6 to 30.9) (Fig 3). The one-, two-, and three-year OS was 50.0%, 50.0%, and

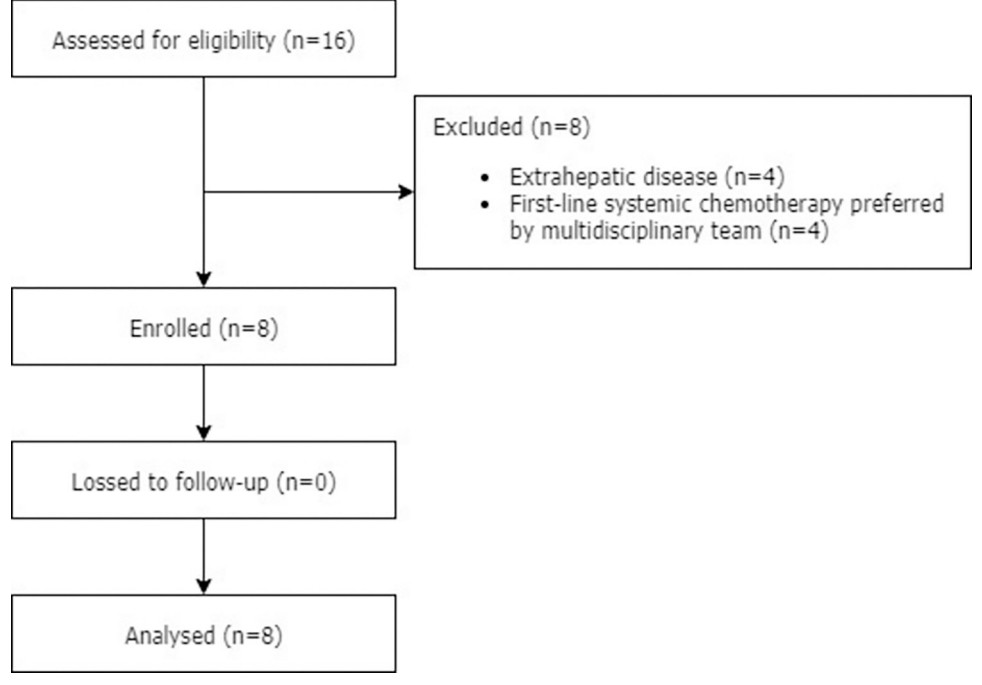

**Fig 1. Patient flow diagram.**

**Table 2. Baseline characteristics.**

| Pt | Gender | Age | Type CRC | Liver metastases Type | No. | Previous therapy[a] Systemic | Local |
|----|--------|-----|----------|------|-----|----------|-------|
| 1 | M | 57 | Sigmoid | Synchr. | > 10 | Bevacizumab/CAPOX; irinotecan/bevacizumab; panitumumab; CAPOX | Segmentectomy S8 |
| 2 | M | 46 | Sigmoid | Synchr. | 2–5 | CAPOX; panitumumab | Metastasectomy S6 & S8, single RFA |
| 3 | M | 64 | Sigmoid | Synchr. | > 10 | Bevacizumab/CAPOX[b]; CAPOX | Metastasectomy S5 & S6, multiple RFAs |
| 4 | F | 57 | Rectum | Synchr. | > 10 | CAPOX[c]; CRT with capecitabine; irinotecan; cetuximab | Metastasectomy S5-7 |
| 5 | M | 55 | Cecum | Synchr. | > 10 | CAPOX; capecitabine | - |
| 6 | M | 51 | Cecum | Synchr. | > 10 | Bevacizumab/CAPOX; FOLFOX | - |
| 7 | F | 49 | Rectum | Metachr. | > 10 | CRT with capecitabine; 5FU/irinotecan/bevacizumab; FOLFOX/ panitumumab | - |
| 8 | F | 68 | Transverse colon | Synchr. | 2–5 | CAPOX | - |

Pt, patient; CRC, colorectal cancer; CRT, chemoradiation therapy; S, liver segment; Synchr., synchronous; Metachr., metachronous; M-PHP, percutaneous hepatic perfusion with melphalan; No., number; RFA, radiofrequency ablation; CAPOX, capecitabine + oxaliplatin; FOLFOX, oxaliplatin + leucovorin + 5FU; -, no previous local therapy.

[a] Besides resection of the primary tumor.

[b] Neoadjuvant treatment.

[c] Induction therapy.

0% respectively. Median PFS was 4.4 months (ranging from 1.1 to 23.6) and median hPFS was 4.5 months (ranging from 1.1 to 23.6). Six out of eight patients received some form of subsequent treatment after progression of disease occurred (Table 3).

**Table 3. Tumor response and survival.**

| Pt | Time between CRLM and 1st M-PHP (mo) | No. procedures | Tumor response | PFS (mo) | hPFS (mo) | OS (mo) | Treatments after PD |
|----|------|------|------|------|------|------|------|
| 1 | 35.1 | 1 | PD[a] | 1.1 | 3.2 | 7.2 | Capecitabine |
| 2 | 28.1 | 2 | PR | 5.7 | 7.1 | 28.7 | FOLFIRI/bevacizumab; capecitabine |
| 3 | 30.0 | 2 | PR | 5.9 | 5.9 | 25.2 | Capecitabine/bevacizumab; panitumumab; irinotecan |
| 4 | 19.3 | 1 | PD[b] | 1.1 | 1.1 | 2.6 | - |
| 5 | 8.3 | 2 | PD[c] | 3.1 | 3.1 | 9.5 | RT[e]; bevacizumab/irinotecan |
| 6 | 8.7 | 2 | SD | 7.3 | 7.3 | 33.3 | BMS-986156/nivolumab[d]; TAS 102 |
| 7 | 9.5 | 2 | SD | 2.9 | 2.9 | 9.1 | - |
| 8 | 32.1 | 2 | SD | 23.6 | 23.6 | 30.9 | Capecitabine |

BOR, best overall response; FOLFIRI, folinic acid (leucovorin) + fluorouracil (5FU) + irinotecan; hPFS, hepatic progression-free survival; mo, months; M-PHP, percutaneous hepatic perfusion with melphalan OS, overall survival; PD, progressive disease; PFS, progression-free survival; PR, partial response; Pt, patient; RT, radiation therapy; SD, stable disease; TAS 102, trifluridine/tipiracil; -, no treatment after PD.

[a] Although liver disease was stable, there was a new lymph node metastasis.

[b] Progression of liver metastases and development of extrahepatic disease.

[c] After the 1st M-PHP, patient showed SD. After the 2nd M-PHP, patient showed PD.

[d] Phase I/II study of BMS-986156 (i.e. a glucocorticoid-induced tumor necrosis factor receptor–related protein) with our without nivolumab.

[e] RT for bone metastases.

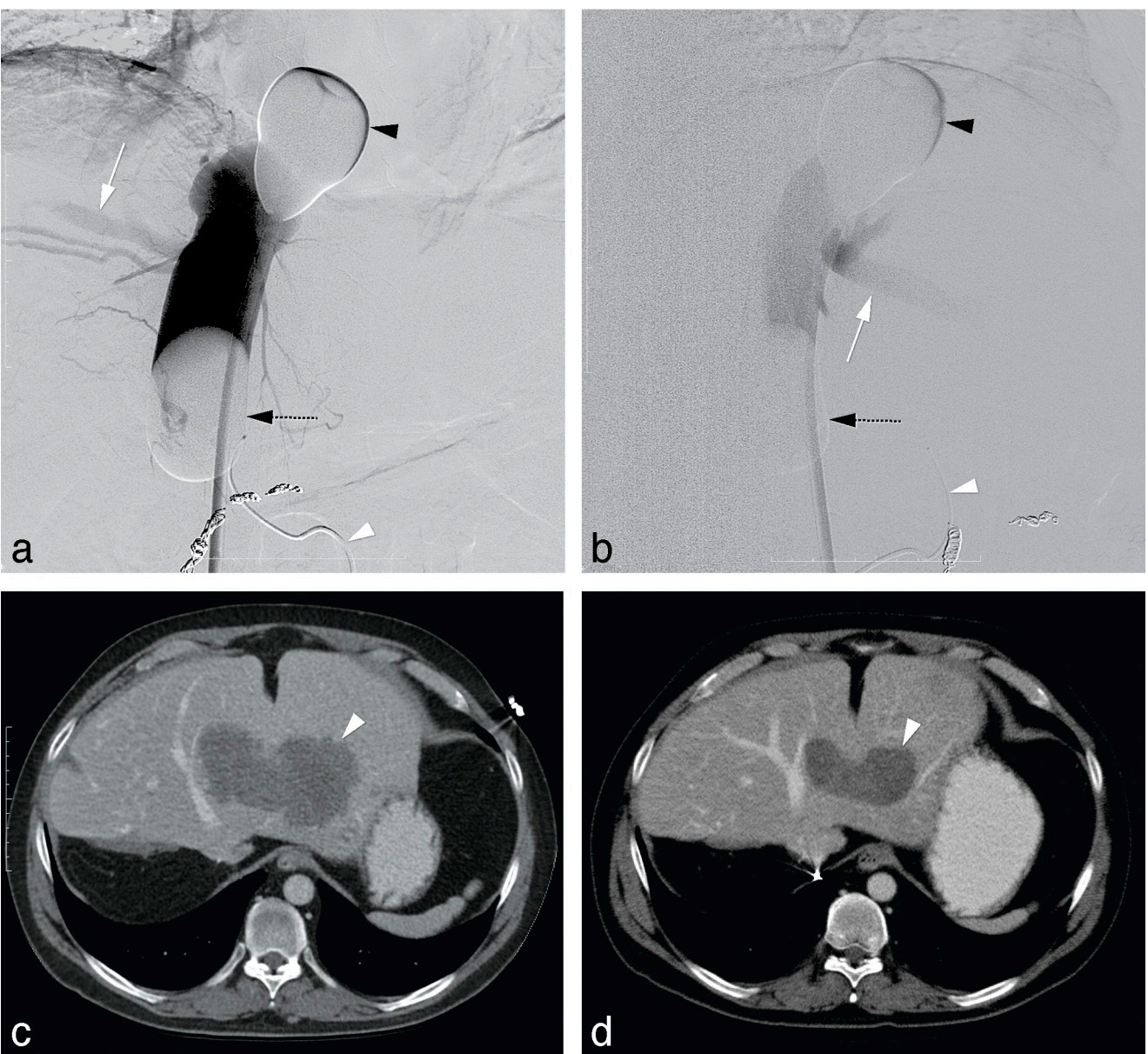

**Fig 2. M-PHP in a 46-year-old male with a solitary liver metastasis from colorectal carcinoma in the left liver lobe after previous radiofrequency ablation in liver segment 2, metastasectomy of liver segment 4, and a right hepatectomy.** (a-b) Postero-anterior and lateral images during venography, performed by manual injection of non-diluted contrast medium through side holes of the double-balloon catheter. The cranial balloon (*black arrowhead*) was inflated at the atriocaval junction and the caudal balloon (*dotted black arrow*) in the infrahepatic portion of the inferior vena cava. Note the opacification of both the right hepatic vein (*white arrow in a*) and middle hepatic vein (*white arrow in b*), while there was no leakage alongside the balloons. A microcatheter (*white arrowhead*) was placed into the hepatic artery proper for the infusion of melphalan. Note also the coils after successful embolization of the right gastric artery and gastroduodenal artery. (c) Axial CT image before treatment showing a solitary hypovascular lesion (*white arrowhead*). (d) Axial CT image after two cycles of M-PHP showing reduction in size of the lesion (*white arrowhead*) corresponding with partial response.

## Safety

No deaths or other serious AEs occurred. All AEs are listed in Table 4. Grade 3/4 thrombocytopenia, anemia, leukocytopenia, and lymphocytopenia was observed in 75.0% (6/8), 37.5% (3/8), 87.5% (7/8) and 100.0% (8/8) of patients, respectively. Grade 4 neutropenia was observed in 50.0% (4/8) of patients. Grade 3 elevation of AST was observed in 25.0% of patients (2/8). The

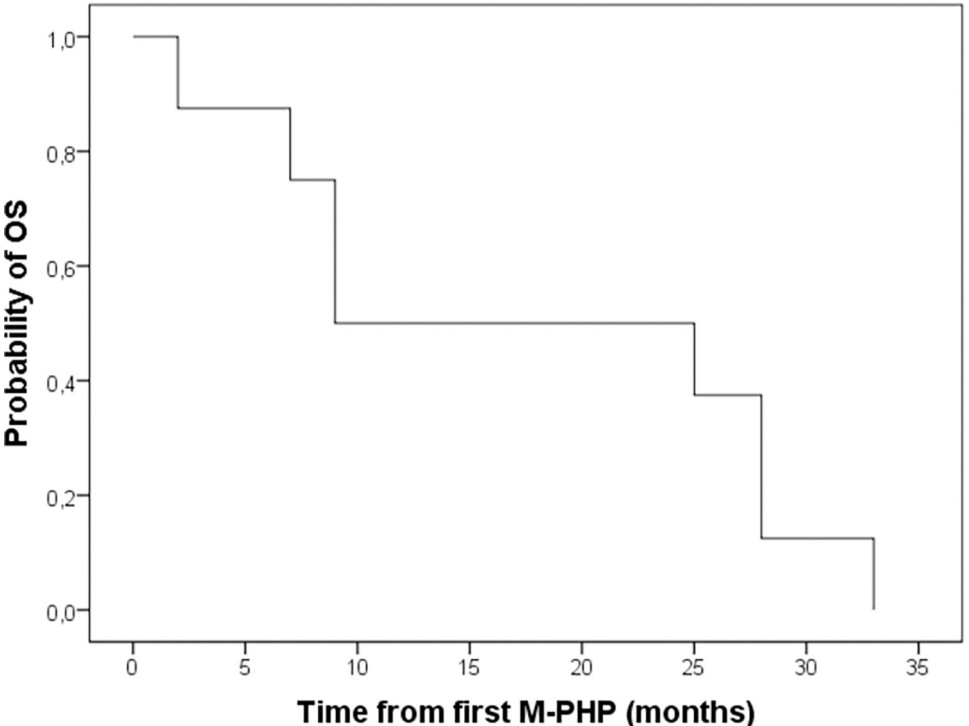

**Fig 3. Kaplan-Meier estimate of OS for all included patients ($n$ = 8).**

most common non-hematologic and non-hepatic AE was grade 1/2 post-procedural fever without any infection focus; this was observed in 50.0% of patients (4/8).

## Discussion

To date, published series including patients with CRLM that were treated with M-PHP have predominantly reported on hemodynamic and metabolic changes, pharmacokinetics and toxicity [23–25]. The current study was designed to prospectively investigate the efficacy and safety of M-PHP in patients with unresectable CRLM. The ORR of 25% and median OS of 17.3 months were lower than expected based on studies on IHP in patients with CRLM [14,15].

Tumor response in our study was unfavorable compared to the prospective study by Rothbarth et al. (ORR 59%) in which 71 patients with CRLM were treated with IHP using a high dose of melphalan [15]. This appears to be largely attributable to a difference in baseline characteristics between both study populations. Whereas 37% of patients in the study by Rothbarth et al. had received a previous treatment for liver metastases (36% systemic therapy and 1% resection) prior to study inclusion, this was 100% in our study population (100% systemic therapy and 50% resection). This was also reflected in a substantial difference in median interval from diagnosis of CRLM to treatment; this was 4 months in the study by Rothbarth et al. versus 23.7 months in the current study. As all patients that were included in our study had already shown disease progression following systemic chemotherapy, it seems plausible that this limited the a priori probability of M-PHP being effective in these patients.

Fluoropyrimidines, i.e. 5-fluorouracil (5-FU) and capecitabin which is the orally administered pro-drug of 5-FU, are the backbone of systemic chemotherapy for metastatic colorectal cancer and are often combined with oxaliplatin (FOLFOX) or irinotecan (FOLFIRI). Although

**Table 4. Adverse events.**

| Patient | 1 | 2 | 3 | 4 | 5 | 6 | 7 | 8 |
|---|---|---|---|---|---|---|---|---|
| Hematologic events | | | | | | | | |
| Thrombocytopenia (gr) | 4 | 3 | 1 | 3 | 2 | 3 | 4[a] | 4 |
| Leukopenia (gr) | 4 | 4 | 4 | 4 | 1 | 4 | 4 | 4 |
| Anemia (gr) | 3 | 2 | 2 | 2 | 2 | 2 | 3[b] | 3[b] |
| Lymphopenia (gr) | 3 | 3 | 3 | 4 | 3 | 4 | 3 | 3 |
| Neutropenia (gr) | 4 | 4 | 4 | n/a | - | 4 | n/a | n/a |
| Non-hematologic events | | | | | | | | |
| Elevated AST (gr) | 2 | 2 | 1 | 3 | 3 | 1 | 2 | - |
| Elevated ALT (gr) | 1 | 1 | - | n/a | n/a | 2 | - | - |
| Elevated bilirubin (gr) | 2 | - | - | n/a | n/a | 2 | - | - |
| Fever, treatment related (gr) | 2 | 1 | - | - | 1 | 1 | - | - |
| Nausea (gr) | - | 1 | 2 | - | 1 | 1 | 2 | - |
| Alopecia (gr) | 1 | - | - | - | - | - | - | 1 |
| Other | c | - | d | - | e | - | - | f |

Note: All patients received a subcutaneous injection of granulocyte-colony stimulating factor within 72h after each M-PHP procedure.

AST, aspartate transaminase; ALT, alanine transaminase; Bili, bilirubin; Gr, grade; n/a, not available; -, no adverse event.

[a] Treated with platelet transfusion.

[b] Treated with red blood cell transfusion.

[c] Haemorrhage groin, treated with a tight pressure dressing.

[d] Peripheral edema due to periprocedural overhydration, treated with diuretics.

[e] Lower urinary tract infection, treated with oral antibiotics.

[f] Aneurysma spurium, successfully treated with a thrombin injection.

percutaneous hepatic perfusion (PHP) with 5-FU has been investigated, currently available hemofiltration systems are only intended for the use of melphalan. Given the disease progression under systemic therapy with fluoropyrimidines in our study population, it seems questionable to treat patients suffering from CRLM with PHP using 5-FU.

We were able to confirm the findings of prior studies that M-PHP is well-tolerated and has an acceptable safety profile [15–18]. This was also the case for one patient with a dihydropyrimidine dehydrogenase (DPD)-deficiency in whom systemic chemotherapy was stopped early due to severe 5-FU-associated toxicity. In a pharmacological study we demonstrated that the mean extraction rate of the GEN 2 hemofiltration system, which is also used in the current study, is 86% [18]. As a result, only a small fraction of melphalan that is administered through the hepatic artery will eventually reach the systemic circulation where it can cause hematologic adverse events. Moreover, all study patients received a G-CSF injection following each M-PHP procedure to further limit systemic side-effects.

This study had some limitations. Most notably, the study was terminated early because of a slow recruitment and therefore the patient number was too low to draw definitive conclusions. Second, the study was a single center study. In relation to this problem, the importance of multi-center recruitment, as performed in other locoregional chemotherapy trials must be emphasized [20,28,29]. Third, there was no control arm.

## Conclusion

We were able to confirm earlier findings that M-PHP is a well-tolerated and safe procedure. The outcomes on tumor response and survival, however, did not meet our expectations and

imply that there currently is no clear role for M-PHP in patients with CRLM outside of clinical trials.

## Supporting information

**S1 Checklist. TREND statement checklist.**
(PDF)

**S1 File. Protocol COLORECTAL PHP_PLOS ONE.**
(PDF)

## Author Contributions

**Conceptualization:** Eleonora M. de Leede, Jaap Vuijk, Christian H. Martini, Fred G. J. Tijl, Ellen Kapiteijn, Alexander L. Vahrmeijer, Mark C. Burgmans.

**Data curation:** T. Susanna Meijer, Eleonora M. de Leede, Alexander L. Vahrmeijer.

**Formal analysis:** T. Susanna Meijer, Ellen Kapiteijn, Mark C. Burgmans.

**Funding acquisition:** Alexander L. Vahrmeijer.

**Investigation:** T. Susanna Meijer, Jan H. N. Dieters, Eleonora M. de Leede, Arian R. van Erkel, Jacob Lutjeboer, Rutger W. van der Meer, Fred G. J. Tijl, Ellen Kapiteijn, Mark C. Burgmans.

**Methodology:** T. Susanna Meijer, Eleonora M. de Leede, Lioe-Fee de Geus-Oei, Jaap Vuijk, Christian H. Martini, Fred G. J. Tijl, Ellen Kapiteijn, Alexander L. Vahrmeijer, Mark C. Burgmans.

**Project administration:** T. Susanna Meijer, Jan H. N. Dieters, Eleonora M. de Leede, Jacob Lutjeboer, Alexander L. Vahrmeijer, Mark C. Burgmans.

**Resources:** Christian H. Martini, Arian R. van Erkel, Jacob Lutjeboer, Rutger W. van der Meer, Fred G. J. Tijl, Alexander L. Vahrmeijer, Mark C. Burgmans.

**Software:** Alexander L. Vahrmeijer.

**Supervision:** Lioe-Fee de Geus-Oei, Ellen Kapiteijn, Alexander L. Vahrmeijer, Mark C. Burgmans.

**Validation:** T. Susanna Meijer, Lioe-Fee de Geus-Oei, Ellen Kapiteijn, Alexander L. Vahrmeijer, Mark C. Burgmans.

**Visualization:** T. Susanna Meijer, Lioe-Fee de Geus-Oei, Ellen Kapiteijn, Alexander L. Vahrmeijer, Mark C. Burgmans.

**Writing – original draft:** T. Susanna Meijer, Jan H. N. Dieters, Lioe-Fee de Geus-Oei, Ellen Kapiteijn, Mark C. Burgmans.

**Writing – review & editing:** T. Susanna Meijer, Eleonora M. de Leede, Lioe-Fee de Geus-Oei, Jaap Vuijk, Christian H. Martini, Arian R. van Erkel, Jacob Lutjeboer, Rutger W. van der Meer, Fred G. J. Tijl, Ellen Kapiteijn, Alexander L. Vahrmeijer, Mark C. Burgmans.

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
