## [Editor Report · Decision Letter 0]

12 Apr 2021

PONE-D-21-02466

Prospective evaluation of percutaneous hepatic perfusion with melphalan as a treatment for unresectable liver metastases from colorectal cancer

PLOS ONE

Dear Dr. Meijer,

Thank you for submitting your manuscript to PLOS ONE. After careful consideration, we feel that it has merit but does not fully meet PLOS ONE’s publication criteria as it currently stands. Therefore, we invite you to submit a revised version of the manuscript that addresses the points raised during the review process.

ACADEMIC EDITOR:

This is a prospective study on a complex locoregional chemotherapy procedure named percutaneous hepatic perfusion. Percutaneous hepatic perfusion was performed with melphalan (high-dosage - approximately 200 mg) and hemofiltration.

The sample size is small (8 patients) and the primary endpoint is response. Considering the small sample size, it is not possible to investigate about the effect of selected variables on response (logistic regression analysis). Secondary endpoints include adverse events that resulted very limited (probably in relation to the use of hemofiltration and the skill of the Authors).

The scientific contribution is important but, in my opinion, the Authors should support with more emphasis the reasons to perform this locoregional chemotherapy procedure.

The indications for M-PHP are still not defined in the international scientific literature.

International guidelines consider locoregional chemotherapy as third line for unresectable CRCLM.

The Authors protocol (as reported in the supporting information file), considers M-PHP for the CRCLM patients for whom “No other standard systemic treatment options (are) available”, but, at same time, include patients “Candidate for neoadjuvant chemotherapy as determined during the multidisciplinary meeting to down size the tumor.” and “Patients (they had) received no, one or more courses of systemic therapy”.

**
*Minor changes*
**

Abstract page 3, line 6: please, added “with hemofiltration”.

Materials and Methods, Procedure details, page 7, line 12: please, clarify if the 16-F double-balloon catheter was positioned percutaneously or by surgical preparation of the femoral (or saphenous) vein.

Table 4: please, precise if patients with grade 3/4 leukopenia/neutropenia received granulocyte colony-stimulating factors.

Table 4: please, re-format line 4 (gr).

In Discussion, the Authors should clarify with more details the reasons for believing and choosing a complex procedure such as M-PHP. In particular the Authors should clarify the reasons to propose percutaneous hepatic perfusion in CRCLM patients not previously submitted to systemic therapy or after failure of the first line.

In Discussion, the Authors should elaborate on the reasons of the very limited adverse events.

In Discussion, as concerning the limitations paragraph, page 12, line 8, please added: “…definitive conclusions. In relation to this problem, the importance of multi-center recruitment, as performed in other locoregional chemotherapy trials […please cite this reference… Mambrini A, Sanguinetti F, Pacetti P, Caudana R, Iacono C, Guglielmi A, Guadagni S, Del Freo A, Fiorentini G, Cantore M. Intra-arterial infusion of 5-fluorouracil, leucovirin, epirubicin, and carboplatin (FLEC regimen) in unresectable pancreatic cancer: results of a ten-year experience. In vivo. 2006; 20: 751-6.], must be emphasized.

We look forward to receiving your revised manuscript.

Kind regards,

Stefano Guadagni

Academic Editor

PLOS ONE
---

## [Author Response · Author response to Decision Letter 0]

1 Jun 2021

Respone to the reviewers

Thank you for reviewing our manuscript. We have appreciated your comments and tried to incorporate them in our manuscript. 

Academic editor:

This is a prospective study on a complex locoregional chemotherapy procedure named percutaneous hepatic perfusion. Percutaneous hepatic perfusion was performed with melphalan (high-dosage - approximately 200 mg) and hemofiltration.

The sample size is small (8 patients) and the primary endpoint is response. Considering the small sample size, it is not possible to investigate about the effect of selected variables on response (logistic regression analysis). Secondary endpoints include adverse events that resulted very limited (probably in relation to the use of hemofiltration and the skill of the Authors).

The scientific contribution is important but, in my opinion, the Authors should support with more emphasis the reasons to perform this locoregional chemotherapy procedure. 

Reply: Although systemic therapy is currently considered the first-line treatment modality in patients with unresectable CRLM, it can be accompanied by serious side-effects. We hypothesised that M-PHP would be a good alternative with limited systemic side-effects for metastatic CRC patients with liver-only disease. It has already been demonstrated that M-PHP is a safe, well-tolerated procedure in patients with liver metastases from ocular melanoma (Cardiovasc Intervent Radiol. 2019;42(6):841–852). Unfortunately, the current study was terminated prematurely due to slow recruitment. Several factors such as the availability of alternative therapies (e.g. systemic chemotherapy or radioembolization) and competing trials with systemic drugs or intra-arterial therapies probably contributed to this slow recruitment. This, together with a tumor response rate and median OS that did not meet our expectations (unfavorable compared to the study by Rothbarth et al. in which patients with CRLM were treated with isolated hepatic perfusion using a high dose of melphalan (Br J Surg. 2003;90(11):1391–1397)) implies that there is currently no clear role for M-PHP in patients with CRLM outside of clinical trials.

The indications for M-PHP are still not defined in the international scientific literature.

International guidelines consider locoregional chemotherapy as third line for unresectable CRCLM.

The Authors protocol (as reported in the supporting information file), considers M-PHP for the CRCLM patients for whom “No other standard systemic treatment options (are) available”, but, at same time, include patients “Candidate for neoadjuvant chemotherapy as determined during the multidisciplinary meeting to down size the tumor.” and “Patients (they had) received no, one or more courses of systemic therapy”.

Reply: It is correct that relatively few exclusion criteria were incorporated in the study protocol whereby most patients with histologically proven, unresectable CRLM were indeed eligible for inclusion. As was shown in Fig. 1, in 4/16 patients that were discussed during a multidisciplinary team meeting prior to possible inclusion, were excluded from participation because first-line systemic therapy was preferred. Table 2 shows that all patients in the current study had received one or more types of systemic therapy before receiving M-PHP. 

The ability to perform PHP may increase the role of hepatic perfusion for treatment of multiple tumor histologies since this procedure allows for sequential treatments to be delivered with lower morbidity. The management of unresectable liver metastases is a significant clinical problem that requires the combined efforts of multiple providers to develop an integrated approach for each patient. Continued evaluation of hepatic perfusion in these patients is necessary so that its role in this integrated approach can be more clearly defined.

Minor changes

Abstract page 3, line 6: please, added “with hemofiltration”.

Reply: We added these words. 

Materials and Methods, Procedure details, page 7, line 12: please, clarify if the 16-F double-balloon catheter was positioned percutaneously or by surgical preparation of the femoral (or saphenous) vein.

Reply: We made alterations accordingly. 

Table 4: please, precise if patients with grade 3/4 leukopenia/neutropenia received granulocyte colony-stimulating factors.

Reply: All patients received a subcutaneous injection of granulocyte-colony stimulating factor within 72h after each M-PHP procedure. We added a sentence in the caption of Table 4.

Table 4: please, re-format line 4 (gr).

Reply: We re-formatted Table 4. 

In Discussion, the Authors should clarify with more details the reasons for believing and choosing a complex procedure such as M-PHP. In particular the Authors should clarify the reasons to propose percutaneous hepatic perfusion in CRCLM patients not previously submitted to systemic therapy or after failure of the first line.

Reply: See also one of the replies to the academic editor. We believe that there is currently no clear role for M-PHP in patients with CRLM outside of clinical trials. In an attempt to clarify this, some changes were made to the manuscript. 

In Discussion, the Authors should elaborate on the reasons of the very limited adverse events.

Reply: In a pharmacological study that was conducted in 2017, de Leede et al. demonstrated that the mean extraction rate of the GEN 2 hemofiltration system is 86%. (Cardiovasc Intervent Radiol. 2017;40:1196–1205). This means that only a small fraction of melphalan that is administered in the hepatic artery will eventually reach the systemic circulation. Additionally, all patients receive a subcutaneous injection with granulocyte colony-stimulating factor after each M-PHP procedure in an attempt to limit any systemic side-effects as much as possible. We made some changes to the Discussion section.

In Discussion, as concerning the limitations paragraph, page 12, line 8, please added: “…definitive conclusions. In relation to this problem, the importance of multi-center recruitment, as performed in other locoregional chemotherapy trials […please cite this reference… Mambrini A, Sanguinetti F, Pacetti P, Caudana R, Iacono C, Guglielmi A, Guadagni S, Del Freo A, Fiorentini G, Cantore M. Intra-arterial infusion of 5-fluorouracil, leucovirin, epirubicin, and carboplatin (FLEC regimen) in unresectable pancreatic cancer: results of a ten-year experience. In vivo. 2006; 20: 751-6.], must be emphasized.

Reply: We made alterations accordingly.

---

## [Decision Letter · Decision Letter 1]

29 Jun 2021

PONE-D-21-02466R1

Prospective evaluation of percutaneous hepatic perfusion with melphalan as a treatment for unresectable liver metastases from colorectal cancer

PLOS ONE

Dear Dr. Meijer,

Thank you for submitting your manuscript to PLOS ONE. After careful consideration, we feel that it has merit but does not fully meet PLOS ONE’s publication criteria as it currently stands. Therefore, we invite you to submit a revised version of the manuscript that addresses the points raised during the review process.

We look forward to receiving your revised manuscript.

Kind regards,

Stefano Guadagni

Academic Editor

PLOS ONE

Journal Requirements:

Additional Editor Comments (if provided):

The revision 1 is good but the paper needs further Minor changes.

Reviewers' comments:

Reviewer's Responses to Questions

**Comments to the Author**

1. If the authors have adequately addressed your comments raised in a previous round of review and you feel that this manuscript is now acceptable for publication, you may indicate that here to bypass the “Comments to the Author” section, enter your conflict of interest statement in the “Confidential to Editor” section, and submit your "Accept" recommendation.

Reviewer #1: All comments have been addressed

Reviewer #2: All comments have been addressed

2. Is the manuscript technically sound, and do the data support the conclusions?

Reviewer #1: No

Reviewer #2: Yes

3. Has the statistical analysis been performed appropriately and rigorously? 

Reviewer #1: No

Reviewer #2: Yes

4. Have the authors made all data underlying the findings in their manuscript fully available?

Reviewer #1: Yes

Reviewer #2: Yes

5. Is the manuscript presented in an intelligible fashion and written in standard English?

Reviewer #1: Yes

Reviewer #2: Yes

6. Review Comments to the Author

Reviewer #1: The manuscript entitled ‘Prospective evaluation of percutaneous hepatic perfusion with melphalan as a treatment for unresectable liver metastases from colorectal cancer’ with the aim to prospectively evaluate the efficacy and safety of M-PHP in patients with unresectable CRLM.

The manuscript can be further improved.

Sample size calculation

Page 9. the write up on the sample size requires further improvement. Information on margin of error, power of study to be stated. The statement of ‘16.5% around the observed proportion’ not clear.

Page 10, what range refers to, min – max or IQR to be clearly stated.

All percentage figures to be at least 1 decimal point.

For Figure 1, more information could be added such as intervention, assessment, outcome variables, duration/period etc.

For Table 2, 3, 4, dash (-) to be denoted in the table footnote.

For Table 3, days to be stated as the initial measurements which then converted to months. This needs to be stated in the methodology. mo to be denoted in table footnote.

In the Reference list, reference number 28 and 29 to conform with the journal format i.e journal name.

Reviewer #2: The paper is very interesting but this is a very complex type of locoregional chemotherapy for hepatic metastases.

In 8 patients the procedure has been performed percutaneously. The sample size is too small to demonstrate that the introduction of a 16F catheter is always possible without complications in all types of femoral arteries (also arteries with a diameter lower than 7 mm).

Minor changes suggested:

The authors should elaborate the previous concepts in Material and Methods and in Discussion.

7. PLOS authors have the option to publish the peer review history of their article (what does this mean?). If published, this will include your full peer review and any attached files.

Reviewer #1: No

Reviewer #2: No

---

## [Author Response · Author response to Decision Letter 1]

1 Nov 2021

Response to the reviewers

Thank you for reviewing our manuscript. We have appreciated your comments and tried to incorporate them in our manuscript. Trying to keep things comprehensive, we responded to the academic editor and reviewer(s) in blue.

Reviewer #1: The manuscript entitled ‘Prospective evaluation of percutaneous hepatic perfusion with melphalan as a treatment for unresectable liver metastases from colorectal cancer’ with the aim to prospectively evaluate the efficacy and safety of M-PHP in patients with unresectable CRLM.

The manuscript can be further improved.

Sample size calculation

Page 9. the write up on the sample size requires further improvement. Information on margin of error, power of study to be stated. The statement of ‘16.5% around the observed proportion’ not clear.

Reply: Unfortunately, no standard power calculation was performed since no clear null-hypothesis could be formulated. We choose a sample size that allows the response percentage to be determined with sufficient accuracy, i.e. with a sufficiently narrow confidence interval. Assuming a true response percentage of 60%, a sample size of 34 patients will yield a two-sided confidence interval of length 0.33 (± 16.5% around the observed proportion). Adjustments were made to our manuscript to clarify this. 

Page 10, what range refers to, min – max or IQR to be clearly stated.

Reply: We changed made alterations accordingly, e.g. see page 10: ‘Median melphalan dose was 220 mg (ranging from 190 to 220) for the first cycle and 220 mg (ranging from range 160 to 220) for the second cycle.’ 

All percentage figures to be at least 1 decimal point.

Reply: We made alterations accordingly. e.g. see page 10: ‘The one-, two-, and three-year OS was 50.0%, 50.0%, and 0% respectively.

For Figure 1, more information could be added such as intervention, assessment, outcome variables, duration/period etc.

Reply: Figure 1 is in our opinion to show the enrolment of patients, and why patients were not eligible for study inclusion. This is all provided in the figure. Elaborating on for instance intervention or outcome variables obtained distracts, in our opinion, from the information currently given in Figure 1. 

For Table 2, 3, 4, dash (-) to be denoted in the table footnote.

Reply: Changes were made accordingly. See Table 2, 3 and 4.

For Table 3, days to be stated as the initial measurements which then converted to months. This needs to be stated in the methodology. mo to be denoted in table footnote.

Reply: Changes were made accordingly. See page 9: ‘Median OS, median PFS, and hPFS were measured in days and subsequently converted into months.’ Also changes were made accordingly in the footnote of Table 3.

In the Reference list, reference number 28 and 29 to conform with the journal format i.e journal name.

Reply: Changes were made accordingly. See reference 28 and 29

Reviewer #2: The paper is very interesting but this is a very complex type of locoregional chemotherapy for hepatic metastases.

In 8 patients the procedure has been performed percutaneously. The sample size is too small to demonstrate that the introduction of a 16F catheter is always possible without complications in all types of femoral arteries (also arteries with a diameter lower than 7 mm).

Minor changes suggested:

The authors should elaborate the previous concepts in Material and Methods and in Discussion.

Reply: This study aimed to evaluate the outcome in patients with colorectal liver metastases treated with percutaneous hepatic perfusion with Melphalan. The safety of this procedure has already been reported by Meijer et al. (Cardiovasc Intervent Radiol. 2019;42(6):841-852.) and is referred to in the last paragraph of the discussion. Until now, arterial access has not been a reason for exclusion of patients for this procedure in our hospital. Therefore, we do not elaborate extensively on this topic in the current paper.

Additional changes:

Figure one has been replace. The first box had a textual error, it said: ‘accessed for eligibility.’ It has now been changed to: assessed for eligibility.’

---

## [Editor Report · Decision Letter 2]

15 Dec 2021

Prospective evaluation of percutaneous hepatic perfusion with melphalan as a treatment for unresectable liver metastases from colorectal cancer

PONE-D-21-02466R2

Dear Dr. Meijer,

We’re pleased to inform you that your manuscript has been judged scientifically suitable for publication and will be formally accepted for publication once it meets all outstanding technical requirements.

Kind regards,

Stefano Guadagni

Academic Editor

PLOS ONE

Additional Editor Comments (optional):

Good revision addressing all my concerns.
---

## [Editor Report · Acceptance letter]

6 Jan 2022

PONE-D-21-02466R2 

Prospective evaluation of percutaneous hepatic perfusion with melphalan as a treatment for unresectable liver metastases from colorectal cancer 

Dear Dr. Meijer:

I'm pleased to inform you that your manuscript has been deemed suitable for publication in PLOS ONE. Congratulations! Your manuscript is now with our production department. 

Kind regards, 

on behalf of

Dr. Stefano Guadagni 

Academic Editor

PLOS ONE